# The Putative Role of the NAC Transcription Factor EjNACL47 in Cell Enlargement of Loquat (*Eriobotrya japonica* Lindl.)

Qian Chen [1,2], Danlong Jing [1,2], Shuming Wang [1,2], Fan Xu [3], Chaoya Bao [3], Ming Luo [3,*] and Qigao Guo [1,2,*]

[1] Key Laboratory of Horticulture Science for Southern Mountains Regions of Ministry of Education, College of Horticulture and Landscape Architecture, Southwest University, Chongqing 400716, China; chenqiansuaige@163.com (Q.C.); jingdanlong@swu.edu.cn (D.J.); wangsm2018@swu.edu.cn (S.W.)

[2] Academy of Agricultural Sciences of Southwest University, State Cultivation Base of Crop Stress Biology for Southern Mountainous Land of Southwest University, Chongqing 400716, China

[3] Key Laboratory of Biotechnology and Crop Quality Improvement, Ministry of Agriculture, Biotechnology Research Center, Southwest University, Chongqing 400716, China; xufanfeiren@163.com (F.X.); cybao0304@126.com (C.B.)

* Correspondence: luomingyuan@swu.edu.cn (M.L.); qgguo75@swu.edu.cn (Q.G.)

**Abstract:** NAC transcription factors (TFs) are plant-specific TFs that play essential roles in plant development; however, the function of NAC TFs in loquat development remains unknown. The natural triploid loquat (*Eriobotrya japonica* Lindl.), Longquan No.1. B355, has larger organs than its corresponding diploid loquat (B2). Here, we cloned an NAC-like TF (EjNACL47 (NAC-like 47)) from the cDNA of triploid loquat B355 flowers. EjNACL47 has a conserved domain of NAC TFs and is homologous to AtNAC47. Transient expression in tobacco leaves revealed that EjNACL47 localized to the nucleus, and yeast-two-hybrid screening confirmed that the C-terminus displayed transcriptional activity. Interestingly, real-time qRT-PCR indicated that the expression levels of *EjNACL47* in leaves and flower organs in triploid loquat (B355) were higher than those in diploid loquat (B2), implying that *EjNACL47* might be associated with the larger organ size in B355. Moreover, *Arabidopsis* lines ectopically expressing *EjNACL47* presented obviously larger leaves, flowers, and siliques than the wild-type variant, suggesting that *EjNACL47* plays a positive role in *Arabidopsis* organ enlargement. These results offer insight into the molecular mechanism of NAC TFs involved in regulating organ size in loquat.

**Keywords:** loquat; NAC transcription factors; organ development; *NAC47*; cell expansion

## 1. Introduction

The development of plant organs to a specific size is the result of the long-term evolution of plant adaption to the environment. At the cytologic level, the growth of plant organs can be divided into two coordinated processes: cell proliferation and expansion [1]. At the molecular level, plant organ size is a complex trait controlled by multiple genes, and its regulation can involve a complex gene network of multiple components and pathways [2,3]. For annual horticultural crops, *fruit weight* (*FW*)2.2 and *FW3.2* regulate cell division during tomato fruit-size evolution [4,5]. *FW11.3* and *physalis organ size 1* modulate fruit size by regulating cell expansion [6,7]. Additionally, a recent study demonstrated that *CsFUL1* modulates cucumber fruit elongation by regulating auxin transport [8]. Although molecular controls associated with the size of annual horticultural crop organs, such as tomato and cucumber, are well known [4–8], knowledge regarding perennial fruit trees is unclear.

Loquat (*Eriobotrya japonica* Lindl.) of the apple subfamily genus belonging to the Rosaceae family is a subtropical evergreen fruit tree native to southern China. Loquat fruit ripens in late spring or early summer [9,10], making it a popular fruit worldwide. However,

because the seeds of loquat fruit are too large and the pulps small, and given the susceptibility to various biotic or abiotic stresses, breeding loquat with excellent agronomic traits is important for loquat breeders. Triploid plants have excellent agronomic traits, including larger organs and stronger resistance to stress [11]. Larger organs, such as leaves, flowers, and fruits, make triploid plants important for breeding. Although fruit thinning [12] and application of plant-growth regulators [13] are widely used to increase fruit size in loquat, the mechanisms of organ-size regulation have not been fully elucidated. Su et al. [14] found that loquat proliferated only a small proportion of cell layers after fruit set and suggested that regulation of cell size would be a more promising aim for further fruit size-related breeding. Further research revealed that cell size is more important to fruit size than cell number in loquat, and that expression of the *EjBZR1* (*BRASSINAZOLE-RESISTANT 1*) gene is negatively correlated with cell and fruit size [15]. However, the molecular mechanisms underlying organ-size regulation in loquat remain poorly understood.

The NAC family of transcription factors (TFs) is one of the largest families of unique plant TFs. Plant *NAC* is derived from the acronym *NAM* (*no apical meristem*) cloned from *Petunia hybrida* in 1996 [16] and *ATAF1/2* and *CUC2* (*cup-shaped cotyledon*) cloned from *Arabidopsis thaliana* [17]. Numerous studies show that *NAC* plays important roles in plant growth and development, response to stress, and other processes [16–23]. Rose (*Rosa hybrida*) *RhNAC100* regulates the expression of genes related to cell elongation [22]. *SlNAP2* plays an important role in regulating leaf senescence and fruit yield in tomatoes (*S. lycopersicum*) [23]. Overexpression of *AdNAC72* in kiwifruit not only enhances *AdMsrB1* expression, but also increases free methionine (Met) and 1-aminocyclopropane-1-carboxylic acid content and ethylene-production rates [18]. Additionally, grapevine (*Vitis vinifera*) *VviNAC33* facilitates the transition from a vegetative to mature phase by inducing leaf degreening and growth cessation [19], and *FaRIF* is a key regulator of strawberry fruit ripening from early developmental stages by controlling abscisic acid biosynthesis and signaling, cell wall degradation and modification, the phenylpropanoid pathway, volatile production, and the balance of aerobic/anaerobic metabolism [20]. Moreover, *Picea wilsonii PwNAC11* plays a dominant role in plants that respond positively to early drought stress [21]. However, relatively few studies have been conducted on *NAC* in relation to loquat. Current studies on loquat *NAC* have mainly focused on loquat fruit postharvest lignification aspects [24,25]. *EjNAC1* could transactivate the gene promoters in loquat and *Arabidopsis* for genes in the lignin biosynthesis pathway. Transient over-expression of *EjNAC1* in tobacco leaves resulted in the accumulation of lignin and induction of the expression of endogenous lignin biosynthesis genes. In conclusion, *EjNAC1* was associated with fruit lignification by activating genes involved in lignin biosynthesis. EjNAC3 trans-activated the lignin biosynthesis-related *EjCAD-like* promoter. Further analysis indicated that EjNAC3 could physically bind to the promoter of the *EjCAD-like* gene. Thus, EjNAC3 is a direct regulator of loquat chilling-induced lignification, via regulations of *EjCAD-like*. However, there are no studies on NAC-related molecular mechanisms associated with organ-size regulation in loquat.

Rauf et al. [26] first characterized *SPEEDY HYPONASTIC GROWTH* (*SHYG*; *AtNAC47*), which is induced by waterlogging (i.e., root submergence), in *Arabidopsis*. Overexpression of *SHYG* in transgenic *Arabidopsis* enhances waterlogging-triggered hyponastic leaf movement and cell expansion in abaxial cells of the basal petiole region, whereas both responses are largely diminished in *shyg*-knockout mutants. Additionally, more than a dozen genes related to cell expansion responded to *SHYG* expression [26]. Because cell expansion might play a key role in organ-size regulation in loquat and the role of *NAC47* in cell expansion has not been further reported, in this study, we isolated an *NAC* gene from the natural triploid of Longquan No.1 loquat (*E. japonica* Lindl.) (B355), which encodes a protein (EjNACL47) highly homologous to *Arabidopsis* AtNAC47. The results also indicated that *EjNACL47* plays a role in cell expansion and organ-size regulation.

## 2. Materials and Methods

### 2.1. Plant Materials

The loquat materials were the natural triploid of Longquan No.1 loquat (*Eriobotrya japonica* Lindl.) B355 and diploid loquat of Longquan No.1 loquat B2, which were preserved in the polyploid loquat resource garden of the Key Laboratory of Fruit Science, Southwest University. The wild-type *Arabidopsis* background used for transformation was Columbia and preserved in our laboratory. The culture condition was 22 °C with a 16 h/8 h light/dark cycle. *Nicotiana benthamiana* was the tobacco variety used for transient expression and was also preserved in our laboratory. The culture condition was 25 °C with a 16 h/8 h light/dark cycle.

### 2.2. EjNACL47 Cloning

Homologous genes of *AtNAC47* were identified from the B355 loquat flower bud transcriptome database (unpublished). Primers were designed according to the selected sequences using SnapGene software (https://www.snapgene.com/, accessed on 10 March 2021). The cDNA B355 loquat flower bud was used as the template to amplify the full-length fragment of the target gene via PrimeSTAR Max premix (2×) (Takara, Shiga, Japan). The reaction conditions were as follows: 95 °C for 5 min, followed by 35 cycles of 95 °C for 30 s, 56 °C for 30 s, 72 °C for 10 s, and 72 °C for 10 min. The primers used for amplification are listed in Supplementary Table S1.

### 2.3. Bioinformatics Analysis

EditSeq (www.dnastar.com, accessed on 30 March 2021.) was used to determine the open reading frame and translated amino acid sequences in order to predict the protein molecular weight and isoelectric point. Chromosome locations of *EjNACL47* were analyzed using BioEdit software to align the sequence against the loquat genome sequence and obtain the chromosome position of *EjNACL47* [27]. We then used Basic Local Alignment Search Tool (https://blast.ncbi.nlm.nih.gov/Blast.cgi, accessed on 30 March 2021) to identify homologous sequences. Homology alignment of EjNACL47 with MdNAC47-like(XP_028946255.1), PmNAC25(XP_008240449.1), PaNAC29-like(XP_021830120.1), PpNAC29(XP_020424848.1), PdNAC29(XP_034226057.1), DzNAC47-like(XP_022742166.1), AtNAC47(NP_187057.2), AtNAC2(NP_188170.1), AtNAC25(NP_564771.1), NAM/CUC2-like(AAB71483.1), and AtRD26(OAO97067.1) was performed using DNAMAN software (https://www.lynnon.com/, accessed on 30 March 2021). Evolutionary analyses were conducted using MEGA7 [28], and evolutionary history was inferred using the neighbor-joining method [29]. The percentage of replicate trees in which the associated taxa clustered together in the bootstrap test (1000 replicates) is shown next to the branches [30]. Evolutionary distances were computed using the number of differences method [31] and are given in units of the number of amino acid differences per sequence.

### 2.4. Real-Time qRT-PCR Analysis

Total RNA was isolated from B2 and B355 loquat material using the plant total RNA extraction kit (Tiangen, Beijing, China) according to manufacturer's instructions. The first strand of cDNA was synthesized from 1 µg of total RNA using SuperScript reverse transcriptase (Takara). Gene-expression levels were measured by qRT-PCR using a Novostar-SYBR supermix kit (NovoProtein, Shanghai, China) on a Bio-Rad CFX96TM machine (Bio-Rad, Hercules, CA, USA), with *EjActin* used as an internal reference. The reaction conditions were as follows: 95 °C for 3 min, followed by 40 cycles of 95 °C for 15 s, 56 °C for 30 s, and 72 °C for 30 s. The program temperature of the melting curve ranged from 65 to 95 °C. Relative expression levels were calculated using the $2^{-\Delta\Delta Ct}$ method [32], and each reaction was performed in triplicate. The primers used are listed in Supplementary Table S2.

## 2.5. Subcellular Localization

To verify its expression in the nucleus, the full-length coding sequence of *EjNACL47* (without the terminating codon) was ligated into the binary vector CaMV35S::GFP (modified from pCAMBIA2300) (Supplementary Table S1). The resultant CaMV35S::EjNACL47-eGFP plasmid was introduced into *Agrobacterium tumefaciens* strain GV3101 and infiltrated into tobacco (*Nicotiana benthamiana*) leaves for transient assays.

Strain GV3101 was activated by YRK (YEB + 50 mg/mL rifampicin + 50 mg/mL kanamycin) on a plate and cultured at 28 °C for 2 days. Single colonies were selected and placed in medium containing 2 mL YRK for overnight culture at 25 °C and shaking at 250 rpm, followed by transfer of 30 μL of the bacterial solution into 25 mL YRK medium (including 10 mM MES and 200 mM acetosyringone) for further culture until reaching an optical density at 600 nm ($OD_{600}$) of ~1.2 to ~1.5. After centrifugation at 5000 rpm for 10 min, the supernatant was discarded, and the bacteria were resuspended in an equal volume of 10 mM $MgCl_2$. After adding 200 mM acetosyringone, the bacteria were incubated in the dark for 3 h. Young leaves of *N. benthamiana* cultured to ~4 weeks of age (4–5 true leaves) and in a good growth state were selected for injection. Before injection, a wound was gently cut in the lower epidermis of the leaf with a needle, and then the bacterial fluid was transferred with a 1 mL syringe and slowly injected into the leaf at the wound. After culturing in the dark for 12 h, the cells were moved to an incubator for normal growth. After 48 h of infiltration, the infiltrated *N. benthamiana* leaves were observed using a confocal laser scanning microscope (SP8; Leica, Wetzlar, Germany).

## 2.6. Transcriptional Activity Assay

The transcription activation assays were performed using the Yeastmaker Yeast Transformation System 2 (Clontech, Mountain View, CA, USA). Different regions of EjNACL47 were cloned into the pGBKT7 vector, and these plasmids and the control plasmid pGBKT7 were introduced into the yeast strain Y2HGold. Transformants were grown on SD medium lacking tryptophan (SD-Trp) and SD medium lacking Trp, histidine (His), and adenine (Ade) (TDO). The primers used for the assays are listed in Supplementary Table S1.

## 2.7. Genetic Transformation in Arabidopsis

Plant-expression vectors containing the target genes were transferred into *Arabidopsis* by flower dip-mediated Agrobacterium GV3101 [33]. The constructed plant-expression vector plGN-35S-EjNACL47-NOS-BE (Supplementary Table S1) was used to transform the Agrobacterium GV3101 strain by the electric shock method. After culturing in YRK medium at 28 °C for 2 days, a single bacterial colony was selected and activated in a test tube. Positive bacterial solution (1 mL) was then drawn from the micropipettor and transferred into 250 mL YRK medium for incubation at 28 °C and 250 rpm until reaching an $OD_{600}$ of ~1.2 to ~1.6. After centrifugation at 5000 rpm for 10 min at 25 °C, the supernatant was discarded, and the bacteria were resuspended in the same volume of osmotic medium (15% sucrose, 0.05% MES, and 0.02% silweet) to form the immersion solution. The siliques and open flowers of *Arabidopsis* were cut off, and the inflorescences were immersed in the immersion solution and removed ~15 s later. Finally, black plastic bags were used to cover the aboveground parts of *Arabidopsis* and placed flat in the dark for ~12 h. The plastic bags were then removed and the plants placed in an incubator for further growth. After 1 week, the plants grew well enough to be disseminated again and then cultured routinely until the seeds were received.

## 2.8. β-Glucuronidase (GUS) Histochemical Identification and Cell-Area Statistics

GUS activity was detected in transgenic plants using 5-bromo-4-chloro-3-indolyl-β-D-glucuronic acid as a substrate [34]. A small amount of the sample to be detected was transferred into an appropriate amount of GUS dye and reacted at 37 °C for 1 h to 2 h. Decolorization was performed using a 75% (*v/v*) alcohol wash three to five times, with the alcohol replaced every 30 min. Photographs were taken using a stereomicroscope (SZX9;

Olympus, Tokyo, Japan), and the lower epidermal cells of *Arabidopsis* leaves were observed and photographed using an upright fluorescence microscope (BX41; Olympus). Cell area was measured using ImageJ software (National Institutes of Health, Bethesda, MD, USA).

## 3. Results

### 3.1. EjNACL47 Expression Is Positively Correlated with Organ Size

Triploid loquat B355 is a natural triploid loquat obtained by screening naturally pollinated seeds of diploid loquat B2. The leaves of B355 loquat were larger than those of the corresponding B2 loquat (Figure 1A). For research convenience, we divided the period from prebloom to early bloom into five stages (a–e). During the process of B355 loquat bud growth, the buds of each stage were larger than those of B2 loquat at the same developmental stage (Figure 1B). Therefore, it is of great significance to clarify the molecular mechanism of organ-size regulation in triploid loquat by clarifying the expression and regulatory mechanisms of related genes in various developmental stages of each organ.

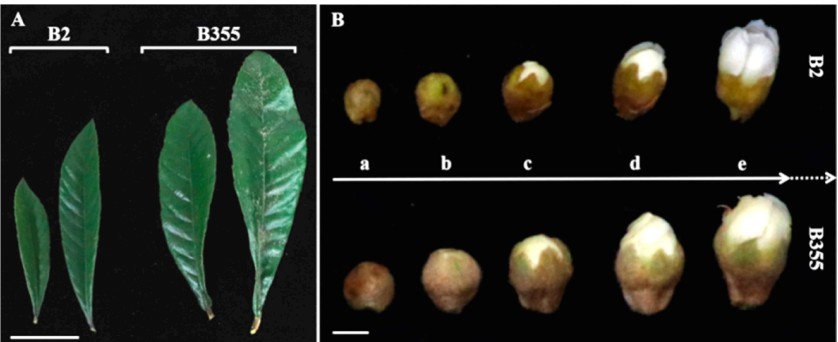

**Figure 1.** Comparison of leaf and flower bud size between B2 and B355 loquat. (**A**) Leaf size difference between B2 and B355 loquat. Scale, 5 cm. The leaves of annual summer shoots and spring shoots are shown on left and right, respectively. (**B**) Differences in the flower buds of B2 and B355 loquat. a–e represent development stages. Scale, 5 mm → represents the development process.

Because *AtNAC47* is related to cell expansion [26], we identified an *AtNAC47* homologous gene (*DN242232_c1_g1*) in loquat by sequence alignment and named it *EjNACL47*. Given that the expression pattern of a gene is usually related to its function, we analyzed expression levels of *EjNACL47* in the leaves and flower buds of B2 and B355 at various developmental stages. The results showed that *EjNACL47* expression increased gradually with the development of flower buds in either B2 or B355, with its expression level in the d- and e-flower buds and young leaves of B355 higher than that in B2 (Figure 2). These results suggested that *EjNACL47* might be involved in regulating organ size.

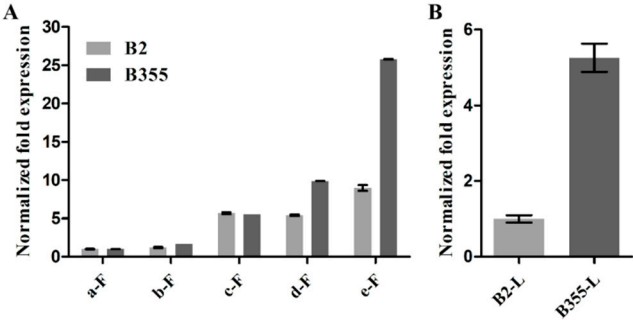

**Figure 2.** Relative expression levels of *EjNACL47*. (**A**) Expression pattern of *EjNACL47* in flower buds (a–e) of B2 and B355 loquat at different developmental stages. (**B**) Expression levels of *EjNACL47* in leaves of B2 and B355 loquat. F: flower buds; L: leaf. Error bars represent the standard deviation of three independent replicates.

### 3.2. EjNACL47 Isolation and Sequence Analysis

To explore the molecular mechanism regulating organ size in B2 and B355, we cloned *EjNACL47* (Supplementary Table S1). The resulting cDNA sequence was 1201 bp in length and encoded a protein of 377 amino acid residues (Supplementary Figure S1), with a predicted molecular weight of 42.0 kDa and an isoelectric point of 8.612 (Supplementary Figure S2A). The chromosome locations of EjNACL47 were identified using BioEdit software to align the sequence against the genome sequences of loquat. The results show that EjNACL47 were located on chromosome 17 (Supplementary Figure S3).

EjNACL47 contains an NAC domain with a sequence common to the NAC family [35] and shares high similarity at the amino acid level with AtNAC47 (Supplementary Figure S4). Compared with other NAC proteins, EjNACL47 harbored a highly conserved N-terminus, whereas the C-terminus differed from those of other NAC proteins (Supplementary Figure S2A). Phylogenetic analysis showed that EjNACL47 was highly homologous to MdNAC47-like protein and demonstrated varying degrees of homology with PmNAC25, PaNAC29-like, PpNAC29, PdNAC29, DzNAC47-like, AtNAC47, AtNAC2, AtNAC25, NAM/CUC2-like, and AtRD26 (Supplementary Figure S2B).

### 3.3. Subcellular Localization of EjNACL47

To examine the subcellular localization of EjNACL47, we constructed the CaMV35S::EjNACL47-eGFP vector (Supplementary Table S1) and introduced it into a transient expression system in *N. benthamiana* leaves using the agrobacterium-mediated method (Supplementary Figure S5). Confocal laser scanning microscopy showed that the GFP signal was exclusively present in the nucleus and overlapped with the fluorescent signal of the nuclear dye 4′,6-diamidino-2-phenylindole (DAPI) (Figure 3A). These results indicated that EjNACL47 localizes to the nucleus.

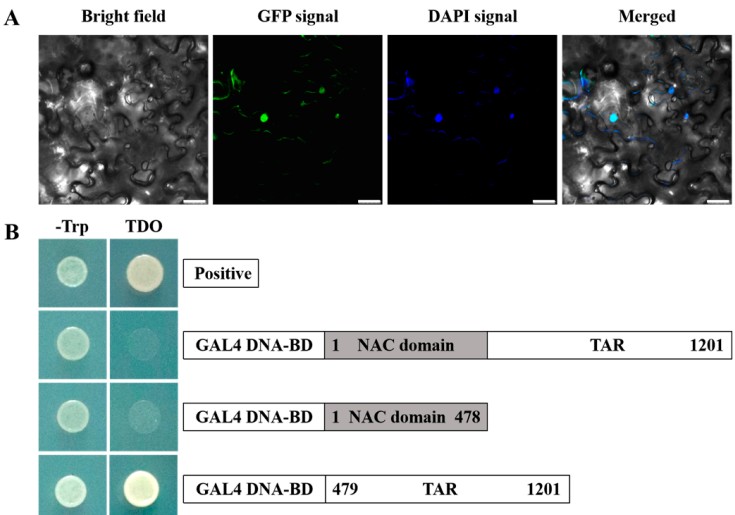

**Figure 3.** EjNACL47 is an NAC TF. (**A**) Subcellular localization of EjNACL47::eGFP. DAPI, 4′,6-diamidino-2-phenylindole dihydrochloride. Merged represents the merged images of bright field, eGFP, and DAPI signals. Scale bar, 25 μm. (**B**) Analysis of EjNACL47 transcriptional activity. TAR, transcriptional activation region; positive, positive control (pGBKT7-p53); BD, pGBKT7 vector; TDO, triple dropout supplements (SD/-Trp-His-Ade).

### 3.4. EjNACL47 Transcriptional Activity

EjNACL47 alignment with other plant NAC proteins showed that EjNACL47 harbored a highly conserved N-terminal NAC domain for DNA binding, although the location of the activation domains remains unknown. We then investigated EjNACL47 transcriptional activity by cloning different regions of the EjNACL47 CDS into the pGBKT7 (BD) vector (Supplementary Table S1). The results showed that yeast cells carrying the BD-transcriptional

activation region (TAR) grew well in SD/-Trp and TDO medium (Figure 3B), indicating that the EjNACL47 TAR exhibited transcriptional activation activity. Conversely, yeast cells carrying the BD-full length or BD-NAC domain only grew on SD/-Trp medium (Figure 3B). These results suggested that the activation domain of EjNACL47 is located at the C-terminus.

### 3.5. EjNACL47 Promotes Organ Enlargement in Arabidopsis

To characterize the function of *EjNACL47*, transgenic *Arabidopsis* plants were generated using the flower-dip strategy (Supplementary Figure S6), in which *EjNACL47* was expressed ectopically (Figure 4). Positive plants were identified by GUS staining [34] (Figure 4A) and DNA amplification (Supplementary Table S1 and Figure 4B), and *EjNACL47* expression was detected by qRT-PCR (Figure 4C). Three transgenic lines (OE-1, -2, and -5) were selected for further analysis.

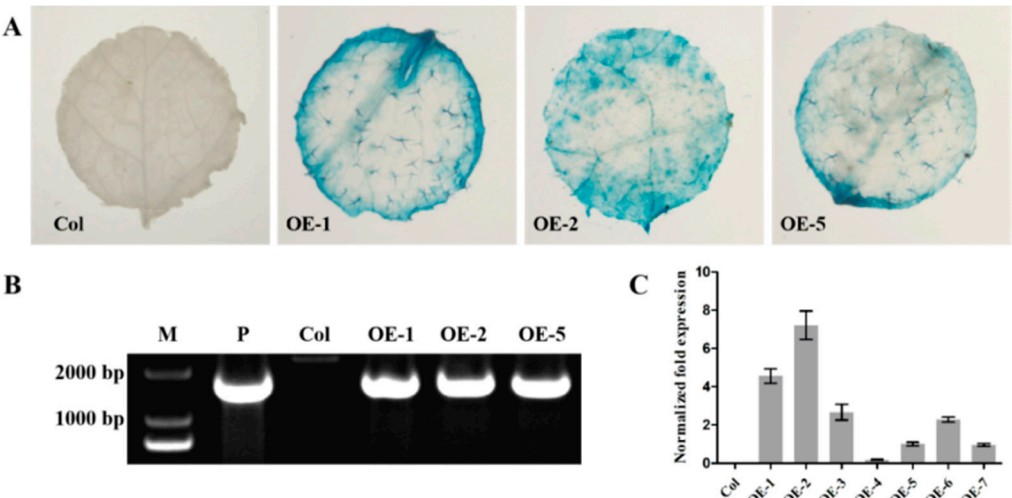

**Figure 4.** Identification of ectopic expression of the *EjNACL47* gene in *Arabidopsis*. (**A**) GUS histochemical identification of transgenic plants. (**B**) PCR identification of transgenic plants; primers are 35S terminal primer and NOS terminal primer. M: marker 2000; P: positive, template is sense EjNACL47 plasmide. (**C**) Relative expression level of *EjNACL47* in the transgenic plants. Error bars represent SD of three independent replicates.

To analyze the effect of ectopic expression of *EjNACL47* on the growth and development of *A. thaliana*, we observed the phenotypes of the T3 generation in the three transgenic lines (OE-1/2/5). The results showed that the petiole of transgenic *Arabidopsis* became longer, and the whole leaf disc became wider than that of the wild type (Figure 5A,C). Additionally, the flower organs of transgenic *Arabidopsis* were significantly increased (Figure 5B), and the silique length of transgenic plants was also longer than that of Col-0 plants (Figure 5D,G). Moreover, observation of root growth by vertical culture revealed that transgenic *Arabidopsis* had longer primary roots than the wild type (Figure 5E,F). The results indicated that ectopic expression of *EjNACL47* promoted the development of *Arabidopsis*, and that the organs of transgenic plants were significantly larger than those of the wild type.

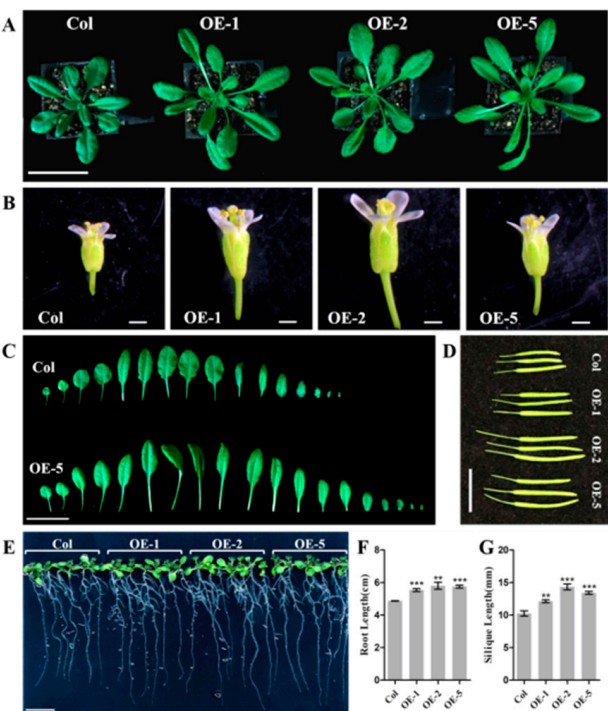

**Figure 5.** Ectopic expression of *EjNACL47* enlarges organ size in *Arabidopsis*. (**A,C**) Phenotypic changes in transgenic *Arabidopsis* at 3 weeks of age. Scale bars, 5 cm and 3 cm, respectively. (**B**) Phenotypic changes in transgenic *Arabidopsis* flowers. Scale bar, 1 mm. (**D,E**) Comparison of fruit silique length and root length between wild-type and transgenic *Arabidopsis*, respectively. Scale bar, 1 cm. (**F,G**) Statistics on root length and silique length of wild-type and transgenic *Arabidopsis*, respectively. Error bars represent the standard deviation from five root-length measurements and 10 silique-length measurements, respectively. ** $p < 0.01$, *** $p < 0.001$.

*3.6. EjNACL47 Promotes Cell Expansion in Arabidopsis by Enhancing the Expression of Expansin and Xyloglucan Endotransgluco Sylase/Hydrolase (XTH) Genes*

To further analyze the specific mechanism of organ enlargement in *Arabidopsis*, we evaluated the lower epidermal cells of transgenic *Arabidopsis* leaves, finding them generally larger than those of the wild type (Figure 6A and Supplementary Figure S7). To determine its significance, we measured the area of the epidermal cells, revealing that the area of transgenic *Arabidopsis* lower epidermal cells was larger than that of the wild type (Figure 6B). Furthermore, RT-PCR analysis showed that the expression of six *expansin* and five *XTH* genes encoding cell-wall-loosening proteins was enhanced in cells overexpressing *EjNACL47* (Figure 6C).

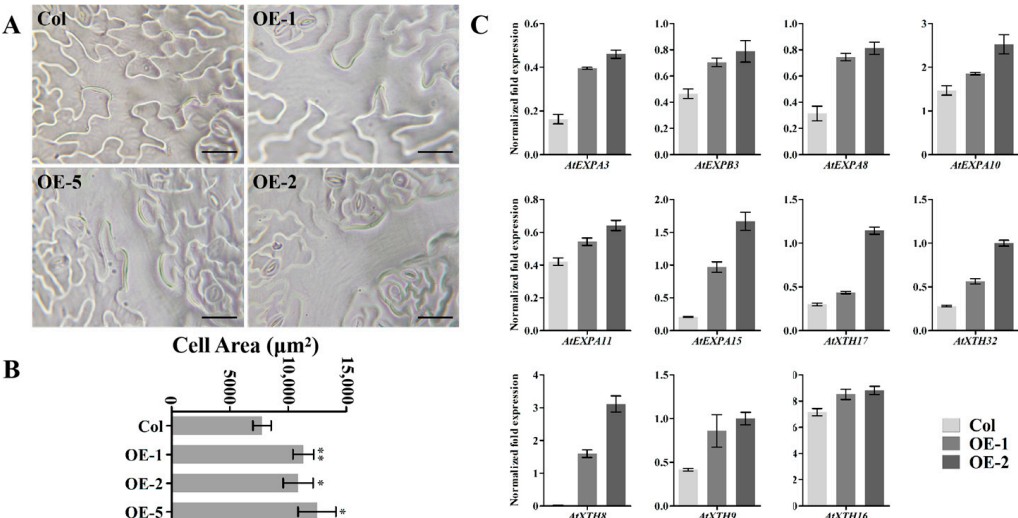

**Figure 6.** Ectopic expression of *EjNACL47* enlarges cell area in *Arabidopsis*. (**A**) Changes in lower epidermal cells of transgenic *Arabidopsis*. Scale bar, 50 μm. (**B**) Statistical analysis of wild-type and transgenic *Arabidopsis* cell area. Error bars represent the standard deviation from 25 cell-area measurements. * $p < 0.05$, ** $p < 0.01$. (**C**) Differentially expressed *expansin* and *XTH* genes in transgenic *Arabidopsis*. FPKM, fragments per kilobase per million.

## 4. Discussion

The 150 amino acids at the N-terminus of NAC TFs are highly conserved and comprise the binding domain, whereas the C-terminal transcriptional regulatory domain of NAC TFs is diverse. The N-terminus is divided into five subdomains (A–E), of which the A subdomain is related to the formation of a functional dimer structure. The B and E subdomains are not highly conserved and are related to the functional diversity of NAC proteins, whereas the C and D subdomains contain nuclear-localization signals and are highly conserved in plants and involved in DNA binding [36,37]. The D subdomain of some NAC proteins contains a highly hydrophobic negative-regulatory region that inhibits transcriptional activation activity [38]. The transcriptional regulatory region is located in the highly variable C-terminus and is capable of either activating [39,40] or inhibiting gene transcription [41–43]. In the present study, we identified EjNACL47 as an NAC TF with typical structural characteristics. Specifically, EjNACL47 localized to the nucleus and exerted transcriptional activation activity from its C-terminus. Notably, transcriptional activity was not present in the full-length protein, indicating that the N-terminal domain might play a regulatory role in transcriptional activation.

*NAC* expression is induced by many environmental factors and plant-developmental stages, with the degree of expression varying in different plants, stages, and organs. In *Arabidopsis*, *NAC1* expression differs between roots, stems, and leaves, with the highest expression observed in roots and relatively low expression in stems and leaves [44]. *CarNAC1* is expressed in various organs of chickpeas, and its expression increases with leaf age and changes during seed development and germination, suggesting a possible role in plant development [45]. The growth of plant organs involves two continuous processes: cell proliferation and expansion. After formation of the organ primordium, cells begin to proliferate, and as the organ grows, the proliferation of cells located at the apex of the primordium stops, and the cells begin to expand [46]. In the present study, the qRT-PCR results showed that *EjNACL47* expression increased gradually with the development of flower buds, with expression levels in B355 loquat leaves and d and e stages flower buds higher than those observed in B2. These findings suggest that *EjNACL47* might play a role in organ-size regulation, most likely through its involvement in cell expansion.

Previous studies report that plants can compensate for impeded plant-cell proliferation by increasing cell size in order to reach the final organ size [47]. The increase in

cell size is the result of increased cytoplasm content and cell-wall loosening, with the expansion of cells inevitably accompanied by an increase in cytoplasmic content [48,49]. The cell wall is wrapped around the cell membrane, which plays a protective role [50], and the increase in cell size necessitates cell-wall loosening to accommodate the increased cytoplasm content. The regulatory factors involved in plant cell-wall loosening are mainly expansin, XTH, and glycoside hydrolase. Expansin breaks the hydrogen bonds between polysaccharides and causes a rearrangement of microfilaments in the cell wall, leading to cell-wall loosening [51]. XTH catalyzes the hydrolysis and reconnection of xyloglucan and regulates cell-wall loosening and remodeling through a mechanism known as "molecular grafting" [52,53]. A previous study showed that Rose (*Rosa hybrida*) *RhNAC100* overexpression in *Arabidopsis* substantially reduced petal size by repressing petal-cell expansion, whereas *RhNAC100* silencing in rose petals significantly increased petal size and promoted cell expansion in the petal abaxial subepidermis. Expression analysis showed that 22 of the 29 cell-expansion-related genes evaluated exhibited changes in expression in *RhNAC100*-silenced rose petals [22]. In the present study, Rauf et al. observed enhanced expression of several *expansin* and *XTH* genes encoding cell-wall-loosening proteins in *SHYG* (*AtNAC47*) overexpressors, but decreased expression in *shyg* mutants, indicating that *AtNAC47* positively regulates the expression of *expansin* and *XTH* genes and cell expansion [26]. Consistently, compared with the wild-type variant, the cell size of *Arabidopsis* overexpressing *EjNACL47* was significantly larger, and the expression of six *expansin* and five *XTH* genes was enhanced in *EjNACL47* overexpressors. We speculated that *EjNACL47* might also promote cell expansion in *Arabidopsis* through direct or indirect regulation of *expansin* and *XTH* genes, ultimately leading to organ-size enlargement in *Arabidopsis*. Furthermore, we found that *EjNACL47* is highly expressed in B355 flower buds and leaves, indicating that *EjNACL47* expression level might play a regulatory role in the organ size of loquat. Recent studies have shown that the expression of *MdNAC047* is enhanced under salt stress, and *MdNAC047* directly activates *MdERF3* (*ethylene response factors gene*) expression by binding to the promoter of *MdERF3* to promote ethylene release and enhance tolerance to salt stress [54]. *Arabidopsis NAC047*/*SHYG* directly or indirectly stimulates local cell expansion through direct activation of *ACC OXIDASE5*(*ACO5*), which encodes a key enzyme of ethylene biosynthesis, constituting an intrinsic ET-SHYG-ACO5 activator loop for rapid petiole cell expansion upon waterlogging [26]. Ethylene, as an important hormone for development and stress response, is involved in regulating two important molecular processes during leaf cell expansion, namely, rapid plasmosome acidification to reduce cell wall hardness and enhance cell wall elongation, and upregulation of *EXPANSIN* genes. In conclusion, we believe that *NAC047* may be involved in different aspects of plant development by regulating different target genes. Whether there are ethylene responsive genes involved in *EjNAC47*-mediated development and stress response needs to be further verified. Taken together, this discovery offers a new understanding of the organ-enlargement problem in triploid loquat and highlights the necessity for further investigation of *EjNACL47*-related mechanisms in organ-size regulation.

**Supplementary Materials:** The following are available online at https://www.mdpi.com/article/10 .3390/horticulturae7090323/s1, Figure S1: Cloning of *EjNACL47*; Figure S2: Sequence analysis of EjNACL47; Figure S3: The chromosome locations of EjNACL47; Figure S4: Phylogenetic analysis of NAC members from loquat and *Arabidopsis*; Figure S5: CaMV35S::EjNACL47-eGFP vector construction; Figure S6: PLGN-35S-EjNACL47-NOS-BE vector construction; Figure S7: Lower epidermal cell changes of transgenic *Arabidopsis*; Table S1: Gene-specific primers used in isolation of *EjNACL47* genes and vector construction; Table S2: Gene-specific primers used in RT-PCR analysis.

**Author Contributions:** Conceptualization, Q.G., M.L. and Q.C.; methodology, Q.C.; software, Q.C.; validation, D.J., S.W. and Q.G.; formal analysis, Q.C.; investigation, Q.C. and C.B.; resources, Q.G. and M.L.; data curation, Q.C. and F.X.; writing—original draft preparation, Q.C.; writing—review and editing, Q.C., D.J., S.W., F.X., C.B., M.L. and Q.G.; visualization, Q.C. and C.B.; supervision, Q.G. and M.L.; project administration, Q.G.; funding acquisition, Q.G. All authors have read and agreed to the published version of the manuscript.

**Funding:** This research was funded by the National Key R&D Program of China: No. 2019YFD1000900, National Nature Science Foundation of China: No. 32102321, Fundamental Research Funds for the Central Universities: XDJK2019AA001 & XDJK2020B058, Innovation Research Group Funds for Chongqing Universities: CXQT19005, Characteristic Fruit Industry and Technology System Innovation Team of Chongqing Agriculture and Rural Affairs Commission: No. 2020[3]01.

**Institutional Review Board Statement:** Not applicable.

**Informed Consent Statement:** Not applicable.

**Data Availability Statement:** Not applicable.

**Conflicts of Interest:** The authors declare no conflict of interest. The funders had no role in the design of the study; in the collection, analyses, or interpretation of data; in the writing of the manuscript; or in the decision to publish the results.

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
