# Peer review of "The Putative Role of the NAC Transcription Factor EjNACL47 in Cell Enlargement of Loquat (Eriobotrya japonica Lindl.)"

_horticulturae, doi:10.3390/horticulturae7090323_

Round 1

Reviewer 1 Report

The work of Chen et al represents a well conducted molecular study on the loquat NAC transcription factor NACL47, discussing its interesting role in promoting organ enlargement in different species. The authors have cloned the EjNACL47 from the triploid loquat flowers, while its transient expression in tobacco confirmed its transcriptional activity. When EjNACL47 was ectopically expressed in Arabidopsis, transgenic lines had larger leaves and flowers that WT plants, supporting the positive role of the gene in organ development.

The study provides interesting findings on the functional role of EjNACL47. I also think this study is well preformed, and no doubt, it represents a substantial amount of work. The techniques used in the study are mostly sound, and the results meaningful. Therefore, I find this work interesting, providing some clues on the involvement of NACL47 in organ development in perennial trees.

My minor suggestions are listed below.

  • Use italics for genes through the entire text.
  • Line 41 is a repetition of line 36, you can avoid it.
  • Lines 64: please explain what RhNAC100 and SLNAP2 are. Are these NAC orthologs in tomato?
  • Line 67: please abbreviate Met.
  • Line 68: in which species VviNA33?
  • Line 69: vegetative-to-mature
  • Line 79: be specific. The gene has this role in loquat.
  • I would suggest to place Figure 3 as Supplementary material (not very exciting), and move supplementary Figure S4 in the main text.

Author Response

Dear Reviewer:

On behalf of my co-authors, we thank you very much for giving us an opportunity to revise our manuscript, we appreciate you very much for your positive and constructive comments and suggestions on our manuscript entitled “The loquat NAC transcription factor EjNACL47 promotes organ enlargement in Arabidopsis”(horticulturae-1360597).

We have studied your comments carefully and have made revision in the paper. We have tried our best to revise our manuscript according to the comments, which we would like to submit for your kind consideration. Here are our point-by-point response:

Question 1: Use italics for genes through the entire text. 

Answer: Thank you for your advice. We have carefully checked our manuscript and used italics for genes.

Question 2: Line 41 is a repetition of line 36, you can avoid it.

Answer: Thank you for your advice. We have modified it.

Question 3: Lines 64: please explain what RhNAC100 and SLNAP2 are. Are these NAC orthologs in tomato?

Answer: Thank you very much for your suggestion. We've added the species of the two genes.

Question 4: Line 67: please abbreviate Met.

Answer: Thank you for your advice. We have added.

Question 5: Line 68: in which species VviNA33?

Answer: Thank you for your advice. We have added.

Question 6: Line 69: vegetative-to-mature.

Answer: Thank you for your advice. We have modified it.

Question 7: Line 79: be specific. The gene has this role in loquat.

Answer: Thank you very much for your suggestion. We have described it more specifically.

Question 8: I would suggest to place Figure 3 as Supplementary material (not very exciting), and move supplementary Figure S4 in the main text.

Answer: Thank you for your advice. We accepted your suggestion and made a replacement.

We would like to express our great appreciation to you for comments on our paper. Looking forward to hearing from you.

Thank you and best regards.

Yours sincerely,

Qigao Guo

Key Laboratory of Horticulture Science for

Southern Mountains Regions of Ministry of Education

College of Horticulture and Landscape Architecture

Southwest University

Chongqing 400716, P. R. China

Ming Luo

Key Laboratory of Biotechnology and Crop Quality Improvement

Ministry of Agriculture/Biotechnology Research Center

Southwest University

Chongqing 400716, P. R. China

Reviewer 2 Report

The manuscript describes the putative function of EjNAC47 based on expression analyses and transformation in Arabidopsis. The study is solid and reasonable to understand the putative function of EjNAC47 in loquat. However the information of NAC gene in apple and pear is missing, although they are representative fruit tree species in Genus. The followings are examples that authors can cite. It would be interesting to discuss the similarity and difference of MdNAC47 and EjNAC47.

Jian‐Ping, et al. "An apple NAC transcription factor enhances salt stress tolerance by modulating the ethylene response." Physiologia plantarum 164.3 (2018): 279-289.    

Ahmad, Mudassar, et al. "Genome wide identification and predicted functional analyses of NAC transcription factors in Asian pears." BMC plant biology 18.1 (2018): 1-15.

Please consider not to use the word Arabidopsis for title.

Keywords are a tool to help researcher find the relevant research. The keywords the authors listed is also used in the title, which is meaningless.

Line 56.  BZR1 (BRASSINAZOLE-RESISTANT 1)?

Line 69. vegetative to mature

Line 74.  As well as you listed like Petunia, tomato and strawberries, there are many studies for NAC family in horticultural plants.

Line 246-252. Authors used only 12 genes for phylogenetic analysis, which is too small, because the number of the NAC genes is more than a hundred in Arabidopsis. Also, I think it is important to include some of apple NAC genes whose putative function has been reported.

Line 348. d-  and e-flower?

Line 368-372. It is better to split the sentence. I suppose the former part of this sentence is the result in this study, but the latter part is from the previous study which used shyg mutants.

Genetic locus of EjNACL47 can be presumed since a draft genome of loquat has been released.Jiang, Shuang, et al. "Chromosome-level genome assembly and annotation of the loquat (Eriobotrya japonica) genome." GigaScience 9.3 (2020): giaa015.

Based on the similarity of the sequence, EjNAC47 is much closer to MdNAC47 than AtNAC47. MdNAC47 regulates ethylene production by binding MdERF3, which is not related to cell expansion. Although, similarity of EjNAC47 and AtNAC47 is not high, their function is similar. Could you explain this in more detail ?

Author Response

Dear Reviewer:

On behalf of my co-authors, we thank you very much for giving us an opportunity to revise our manuscript, we appreciate you very much for your positive and constructive comments and suggestions on our manuscript entitled “The loquat NAC transcription factor EjNACL47 promotes organ enlargement in Arabidopsis”(horticulturae-1360597).

We have studied your comments carefully and have made revision in the paper. We have tried our best to revise our manuscript according to the comments, which we would like to submit for your kind consideration. Here are our point-by-point response:

Question 1: Please consider not to use the word Arabidopsis for title.

AnswerThank you very much for your suggestion. We have considered not to use the word Arabidopsis for title, but the current results on regulating organ size are obtained through ectopic expression of Arabidopsis. Therefore, we hope to use Arabidopsis for title to enhance the credibility of our paper, but we have taken your suggestion into consideration in future research.

Question 2: Keywords are a tool to help researcher find the relevant research. The keywords the authors listed is also used in the title, which is meaningless.

AnswerThank you for your suggestion. We have modified keywords.

Question 3: Line 56. BZR1 (BRASSINAZOLE-RESISTANT 1)?

Answer: Thank you for your advice. We have added it.

Question 4: Line 69. vegetative to mature

AnswerThank you for your suggestion. We have modified it.

Question 5: Line 74. As well as you listed like Petunia, tomato and strawberries, there are many studies for NAC family in horticultural plants.

AnswerThank you very much for your suggestion. In fact our main purpose was to show that NAC has been poorly studied in loquat, so we have redescribed the sentence.

Question 6: Line 246-252. Authors used only 12 genes for phylogenetic analysis, which is too small, because the number of the NAC genes is more than a hundred in Arabidopsis. Also, I think it is important to include some of apple NAC genes whose putative function has been reported.

Answer: Thank you very much for your advice. As we described in the introduction, we are using homology-based cloning of the gene, so our phylogenetic analysis is mainly to verify that the protein was highly homologous to AtNAC47, to continue our further experiments to verify whether EjNACL47 can also involved in regulation of cell expansion. Our original intention was to solve the problem of whether EjNACL47 is involved in regulating cell expansion. Based on this, we only selected 12 genes highly homologous to this protein in the comparison results. But we believe that your comments have played an important role in exploring the potential function of EjNACL47. We will focus on the progress of apple research in future research.

Question 7: Line 348. d-  and e-flower?

AnswerThank you for your suggestion. We have modified it.

Question 8: Line 368-372. It is better to split the sentence. I suppose the former part of this sentence is the result in this study, but the latter part is from the previous study which used shyg mutants.

Answer: Thank you for your advice. We have modified it. 

Question 9: Genetic locus of EjNACL47 can be presumed since a draft genome of loquat has been released.Jiang, Shuang, et al. "Chromosome-level genome assembly and annotation of the loquat (Eriobotrya japonica) genome." GigaScience 9.3 (2020): giaa015.

Answer: Thank you for your suggestion. We have added.

Question 10: Based on the similarity of the sequence, EjNAC47 is much closer to MdNAC47 than AtNAC47. MdNAC47 regulates ethylene production by binding MdERF3, which is not related to cell expansion. Although, similarity of EjNAC47 and AtNAC47 is not high, their function is similar. Could you explain this in more detail ?

Answer: Thank you very much for your advice. We've explained this in more detail in the discussion section, please refer to the last paragraph in the discussion section.

We would like to express our great appreciation to you for comments on our paper. Looking forward to hearing from you.

Thank you and best regards.

Yours sincerely,

Qigao Guo

Key Laboratory of Horticulture Science for

Southern Mountains Regions of Ministry of Education

College of Horticulture and Landscape Architecture

Southwest University

Chongqing 400716, P. R. China

Ming Luo

Key Laboratory of Biotechnology and Crop Quality Improvement

Ministry of Agriculture/Biotechnology Research Center

Southwest University

Chongqing 400716, P. R. China

Round 2

Reviewer 2 Report

I think a title is very important to attract researchers to read this manuscript. This title makes readers imagine this study is focusing on Arabidopsis that is not suitable for Horticulturae. Because the loquat is miner species, few readers recognize this word as a species name. The following is just example. I suggest authors to reconsider the title on their own.

“The putative role of the NAC transcription factor EjNACL47 in cell enlargement of Eriobotrya japonica Lindl.”

Authors response: Thank you very much for your advice. As we described in the introduction, we are using homology-based cloning of the gene, so our phylogenetic analysis is mainly to verify that the protein was highly homologous to AtNAC47, to continue our further experiments to verify whether EjNACL47 can also involved in regulation of cell expansion. Our original intention was to solve the problem of whether EjNACL47 is involved in regulating cell expansion. Based on this, we only selected 12 genes highly homologous to this protein in the comparison results. But we believe that your comments have played an important role in exploring the potential function of EjNACL47. We will focus on the progress of apple research in future research.

Comment: I wonder there might be some NAC genes in arabidopsis more similar to EjNACL47 rather than AtNAC47. The only way to clarify the similarity of NAC genes is to provide a phylogenetic analysis using a large numbers of NAC genes as many other studies relevant to NAC genes show. At least authors can provide this type of analysis in supplementary files. Authors added the relationship sof ethylene biosynthesis and cell enlargement, which is reasonable. But there is no explanation about similarity of EjNACL47 and AtNAC47.

Author Response

Dear Reviewer,

On behalf of my co-authors, we appreciate you very much for your positive and constructive comments and suggestions on our manuscript entitled “The loquat NAC transcription factor EjNACL47 promotes organ enlargement in Arabidopsis”(horticulturae-1360597). Thank you for your patience.

We have studied your comments carefully and have made revision in the paper. We have tried our best to revise our manuscript according to the comments,  which we would like to submit for your kind consideration. Here are our point-by-point response:

Question 1: I think a title is very important to attract researchers to read this manuscript. This title makes readers imagine this study is focusing on Arabidopsis that is not suitable for Horticulturae. Because the loquat is miner species, few readers recognize this word as a species name. The following is just example. I suggest authors to reconsider the title on their own.

“The putative role of the NAC transcription factor EjNACL47 in cell enlargement of Eriobotrya japonica Lindl.”

AnswerThank you very much for your suggestion. According to your sincere suggestion, we have changed the title to “The putative role of the NAC transcription factor EjNACL47 in cell enlargement of loquat (Eriobotrya japonica Lindl.)”.

Question 2: I wonder there might be some NAC genes in arabidopsis more similar to EjNACL47 rather than AtNAC47. The only way to clarify the similarity of NAC genes is to provide a phylogenetic analysis using a large numbers of NAC genes as many other studies relevant to NAC genes show. At least authors can provide this type of analysis in supplementary files. Authors added the relationship sof ethylene biosynthesis and cell enlargement, which is reasonable. But there is no explanation about similarity of EjNACL47 and AtNAC47.

AnswerThank you very much for your patient analysis and suggestions.We have added a supplementary figure (Supplementary Figure S4) to explain that the EjNACL47 gene has the highest similarity to AtNAC47 in the NAC  genes of Arabidopsis.

We would like to express our great appreciation to you for comments on our paper. Looking forward to hearing from you.

Thank you and best regards.

Yours sincerely,

Qigao Guo

Key Laboratory of Horticulture Science for

Southern Mountains Regions of Ministry of Education

College of Horticulture and Landscape Architecture

Southwest University

Chongqing 400716, P. R. China

Ming Luo

Key Laboratory of Biotechnology and Crop Quality Improvement

Ministry of Agriculture/Biotechnology Research Center

Southwest University

Chongqing 400716, P. R. China

This manuscript is a resubmission of an earlier submission. The following is a list of the peer review reports and author responses from that submission.

Round 1

Reviewer 1 Report

In the manuscript, Chen et al. isolated the gene of a transcription factor (EjNACL47) in loquat and ectopically expressed it in Arabidopsis, the results indicated that EjNACL47 play a role in enlargement of several vegetative and reproductive organs. It is helpful for understanding the role of NAC transcription factors in plant growth and development. Since research about NAC transcription factors in horticultural plants is rare, the results are novel and interesting. Nevertheless, there are still several points need to be clarified or addressed. 

  1. The language need to be polished carefully.
  2. In the “Introduction” and “Discussion” part, the authors need to address the research about NAC transcription factors as much detail as possible. Especially in the “Discussion” part, the authors need to speculate the possible mechanism of EjNAC47 transcription factors based on achieved results and reports about NAC transcription factors in different plant species. The current introduction and discussion writing is somehow simple and arbitrary.
  3. In “Material and Methods” part, line 118, line 121, line 124, full name of “Km”, “AS” and “As” need to be addressed. Is “AS” and “As” same? Line 143-144, the composition of YRK is unnecessary to be repeated since it has been addressed in line 118.
  4. Line 130, “aconfocal” should be “a confocal”. Line 146, “Pipetting gun” should be ”micropipettor”? Line 160, “changed” should be “replaced”. Line 162, “positive” should be “upright”.
  5. In Figure 1, it is noted that bud of B355 is larger than B2 at Stage b and Stage c. However, in Figure 2, the expression level of EjNAC47 in B355 at Stage b and Stage c is not significantly higher than that in B2. It needs to be explained in the discussion. In Figure 2, the leaf age also needs to be addressed. Are they growing or mature leaves?
  6. In Figure 3, the text size in the figure is too small to distinguish.
  7. Line 222-224, “represen” should be “represent”. Line 242, “determined” should be “identified”.
  8. The figure legend of Figure 5 should be appropriately consolidated and simplified. At the end of the figure legend, “*** means p<0.0001” should be “*** means p<0.001”.